# Taste Receptors beyond Taste Buds

**DOI:** 10.3390/ijms23179677

**Published:** 2022-08-26

**Authors:** Su Young Ki, Yong Taek Jeong

**Affiliations:** 1Department of Pharmacology, Korea University College of Medicine, Seoul 02841, Korea; 2BK21 Graduate Program, Department of Biomedical Sciences, Korea University College of Medicine, Seoul 02841, Korea

**Keywords:** taste, taste receptors, ectopic expression, tuft cells, G protein-coupled receptors

## Abstract

Taste receptors are responsible for detecting their ligands not only in taste receptor cells (TRCs) but also in non-gustatory organs. For several decades, many research groups have accumulated evidence for such “ectopic” expression of taste receptors. More recently, some of the physiologic functions (apart from taste) of these ectopic taste receptors have been identified. Here, we summarize our current understanding of these ectopic taste receptors across multiple organs. With a particular focus on the specialized epithelial cells called tuft cells, which are now considered siblings of type II TRCs, we divide the ectopic expression of taste receptors into two categories: taste receptors in TRC-like cells outside taste buds and taste receptors with surprising ectopic expression in completely different cell types.

## 1. Introduction

The chief function of chemosensory receptors is to recognize exogenous compounds, convert their chemical signals into biologic responses, and generate chemosensory information. Therefore, primary chemosensory tissues, such as the olfactory epithelium and the taste buds, express olfactory and taste receptors, respectively. Chemosensory receptors are also expressed, however, in organs unrelated to sensory perception. This expression was initially considered misregulation and was thus dubbed “ectopic expression”. Since then, new evidence has accumulated suggesting these ectopic chemosensory receptors play other roles. This review focuses mainly on the function of taste receptors expressed outside of taste buds. First, we briefly describe the history of taste receptor research. Then, we describe taste receptor-coupled signaling pathways before finally exploring ectopic taste receptor expression in TRC-like cells outside taste buds as well as genuine ectopic expression. This review paper only refers taste receptors, the subclass of G protein-coupled receptors superfamily, as “taste receptors”, excluding the ion channels involved in the transduction of salty and sour taste.

## 2. Classical View on Taste Receptors

### 2.1. Discovery of Taste Receptors

Taste receptors were first identified via a subtractive cDNA library screen performed on rodent taste bud-containing tissues [1,2]. A similar strategy had already proven successful in the discovery of the olfactory receptors (ORs) and vomeronasal receptors (VRs) in the olfactory mucosa [3,4]. The first taste receptor screen identified TR1 and TR2, later renamed Tas1r1 and Tas1r2, respectively [5]. Sequence analysis of Tas1r1 and Tas1r2 implied their membership in the GPCR superfamily as siblings of the ORs and metabotropic glutamate receptors (mGluRs) [2]. Further screening revealed a group of sequences similar to Tas1r1 and Tas1r2, and these were dubbed Tas2rs [5].

Meanwhile, Tas1r3 was discovered via a different approach [6,7,8,9,10]. Mouse geneticists had already mapped the *Sac* locus on the tip of mouse chromosome 4 as a determinant of behavioral preference toward sweet taste stimuli [11]. Then, six independent groups suggested that a single Gpr70-encoding gene near the *Sac* locus encodes a novel taste receptor family [6,7,8,9,10,12]. Genetic rescue experiments that introduced a taster allele of Tas1r3 into homozygous non-taster animals finally proved this to be true.

Together, three Tas1rs and 25 (in humans) or 35 (in mice) Tas2rs have been identified. It is now widely accepted that Tas1r3 acts as a co-receptor for sweet and umami taste receptors in combination with Tas1r2 or Tas1r1 [12,13], respectively. Tas2rs, in contrast, function as bitter taste receptors [14,15].

### 2.2. Signal Transduction Pathways for Taste Receptors

After taste receptors were identified at the molecular level, amino acid-level sequence comparisons of taste receptors indicated that they possess seven transmembrane domains [1,2] and suggested they likely function as GPCRs. First, sweet, bitter, and umami tastants trigger intracellular increase in Ca^2+^ [16] or decrease in cyclic adenosine monophosphate (cAMP) concentrations [17,18] in type II TRCs. Second, type II TRCs express molecular players for GPCR pathways such as Gβ3 [19], Gγ13 [20], α-gustducin (GNAT3) [18,21,22,23], phospholipase Cβ_2_ (PLCβ_2_) [21], and transient receptor potential M5 (TRPM5) [21,24]. Lastly, in vivo mouse studies of genetic knockouts confirm their significance in the cellular responses to tastants [18,21,23].

Despite being activated by different types of tastants, Tas1rs and Tas2rs couple to the same intracellular signaling pathway via a heterotrimeric G protein [21]. When the receptors are activated by bitter or sweet compounds, the α subunit GNAT3 dissociates from its β and γ subunit partners, which then activate downstream signaling via PLCβ_2_ and the inositol 1,4,5-trisphosphate (IP_3_) pathway. These trigger cytoplasmic Ca^2+^ release [19,20]. IP_3_ triggers Ca^2+^ release from the endoplasmic reticulum via the IP_3_ receptor [25]. The Gα subunit activates phosphodiesterases, which reduce cAMP levels [26]. This reduces protein kinase A (PKA) activation. Without this inhibition [27], PKA prevents the IP_3_ pathway from inducing further Ca^2+^ efflux from the endoplasmic reticulum [28]. Once released, cytoplasmic Ca^2+^ ions open the TRPM5 channel [29,30,31], resulting in Na^+^ influx [32]. This triggers the opening of the voltage-gated Na^+^ channels that mediate action potential generation [33]. Action potentials trigger the release of the neurotransmitter ATP via a non-vesicular mechanism mediated by a heteromeric complex of calcium homeostasis modulator 1 (CALHM1) [34] and CALHM3 [35]. The secreted ATP then activates taste cells and sensory fiber receptors, leading to the transmission of taste sensation to the central nervous system [36,37].

## 3. Categorizing Ectopic Taste Receptor Expression

### 3.1. Taste Receptors Expressed in TRC-Like Cells Outside Taste Buds

Tuft cells are a rare population of epithelial cells named for their characteristic morphology, including a specialized tubulovesicular system and a “tuft” of long, blunt microvilli projecting from a narrowed apical face [38]. Tuft cells are broadly distributed across the epithelial mucosa, such as the oral cavity [39], alimentary tract [40,41,42,43,44], upper and lower respiratory tracts [45,46,47,48], and even the thymus [49,50]. Depending on their location, tuft cells can also be referred to as brush cells [51,52] or solitary chemosensory cells (SCCs) [39,45,46,53,54]. Recently, tuft cells were divided into two groups [43,47]. Tuft-1 cells share very similar transcriptomic features with type II TRCs in taste buds, whereas tuft-2 cells show enrichment for immune-related genes [43,55]. Taste receptors and their downstream effectors are generally expressed in tuft-1 cells. This classification is valid across many organs, such as the intestines [43], the airways [47], and the thymus [49]. Further study of the tuft cells in other organs will improve these sub-classifications. Similar morphology, gene expression profiles, and developmental pathway [56,57,58] suggest that tuft cells are TRC-like cells. Therefore, we need to discriminate the expression of taste receptors in TRC-like cells from in other types of cells.

#### 3.1.1. Intestinal Tuft Cells

Most studies of tuft cells have focused on those in the intestine. Intestinal tuft cells are crucial for defense against microbial challenges because they detect microbe-derived chemicals in the intestinal lumen. Upon activation, these tuft cells secrete interleukin 25 (IL-25) to drive the activation of type 2 innate lymphoid cells 2 (ILC2) [40,42,44,55,58]. Multiple research groups found that intestinal tuft cells exhibit a transcriptional repertoire similar to type II TRCs. The master transcription factor Pou2f3 is necessary for the generation of both intestinal tuft cells and type II TRCs [56,57,58]. Both cell types also share the downstream effectors required for taste receptor signaling. Even before the expression of taste receptors was discovered in intestinal tuft cells, their expression of the taste-specific GPCR signal transducers GNAT3, PLCβ_2_, and TRPM5 had already been reported, implicating them in chemosensation [44]. Later, transcripts encoding taste receptors were sporadically detected by RT-PCR [59]. More recently, single-cell RNA sequencing (scRNA-seq) experiments have also revealed taste receptor expression in intestinal tuft cells [43]. Tas1r3 seems to be expressed alone without its conventional binding partners, Tas1r1 or Tas1r2 [43]. Although bitter receptor expression was too low for detection in multiple single-cell data [41,43], RT-PCR and genetic lineage tracing experiments support their expression in intestinal tuft cells [59,60]. This discrepancy may be an artifact of the limited coverage of single-cell RNA sequencing. Alternatively, given that tuft cell expansion depends on the microbiome in such a way that basal expression levels of bitter receptors would presumably remain low in the normal state, it is possible that the discrepancy is due to differences in the microbiomes of various mouse cohorts. It also remains unclear whether Tas1r3 and bitter receptors are expressed in identical cell populations.

What about the function of taste receptors in intestinal tuft cells? The simplest hypothesis suggests that taste receptors detect various ligands inside the gut lumen. For example, bitter receptors offer appropriate molecular machinery for activating defensive responses to infection by the parasitic helminth Trichinella spiralis [59]. T. spiralis-derived compounds evoke intracellular Ca^2+^ responses in intestinal tuft cells, causing them to release IL-25. They then undergo hyperplastic expansion via a tuft cell-ILC2 circuit. T. spiralis infection increases the expression of several bitter receptors, including the salicin-responsive receptor Tas2r143. Salicin stimulation recapitulated tuft cell secretion of IL-25. In addition, stimulation with allyl isothiocyanate (AITC), which irreversibly antagonizes bitter receptors via an unknown mechanism, impaired IL-25 secretion. These results suggest bitter receptors can play a role in sensing unidentified parasite-derived molecules.

In contrast, Tas1r3, the most abundant taste receptor in the intestine, seems to have another function. Tas1r3-deficient mice have fewer intestinal tuft cells than wild-type control mice and mice with mutations in the chemosensory genes GNAT3 and PLCβ_2_ [61]. The cellular composition of the other types of intestinal epithelial cells, however, remained unaltered. Treatment of Tas1r3 knockout (KO) mice with IL-13 or succinate induced differentiation toward intestinal tuft cells, increasing the number of tuft cells to levels comparable to that of controls. This implies a role for Tas1r3 in the regulation of epithelial homeostasis independent from the canonical chemosensory and ILC2 pathways.

#### 3.1.2. Respiratory Tuft Cells

The respiratory mucosa is another tissue with its own microbiome. Like the intestinal tuft cells, respiratory tuft cells receive microbe-derived molecules to regulate the airway immune system. In the upper respiratory mucosa, TRPM5-expressing SCCs are scattered across the sinonasal surface [54]. These SCCs are distinct from the conventional chemosensory cells in the olfactory mucosa. Pou2f3-deficient mice lack TRPM5 SCCs in the nasal cavity, implying that they follow a similar developmental program as tuft cells in other regions. Transcripts for bitter receptors are expressed in the nasal mucosa, and their expression is regulated by sinonasal infection [62]. Bitter tastants and *Pseudomonas aeruginosa* (*P. aeruginosa*)-derived quorum-sensing molecules such as LasI (3-oxo-C12-HSL) and EsaI (3-oxo-C6-HSL/C6-HSL) evoke Ca^2+^ responses in sinonasal SCCs [53]. Sinonasal SCC activation triggers the release of acetylcholine, which transiently suppresses spontaneous breathing events via nociceptive neurons in the area surrounding the SCCs [46]. Moreover, SCC activation also induces nasal inflammation via substance P, resulting in plasma extravasation and mast cell degranulation. Genetic loss of Gnat3 abolished all these phenomena, implicating the canonical taste receptor transduction pathway [53]. Sinonasal SCCs in humans express bitter receptors like their counterparts in mice, but human sinonasal SCCs also express Tas1r3 [45]. Interestingly, the function of Tas1r3 in these SCCs opposes that of the bitter receptors [45]. Glucose-mediated activation of Tas1r3 was found to alleviate cellular responses to the bitter tastant denatonium, and treatment with the specific Tas1r3 antagonist lactisole was found to disinhibit this inhibitory effect of glucose on the denatonium response.

In the lower respiratory tract, tracheal tuft cells are responsible for neurogenic inflammation induced by bacterial infection [52]. LasI not only activates intracellular Ca^2+^ responses in tracheal tuft cells, but it also induces extravasation and recruits neutrophils via secretion of calcitonin gene-related peptide and substance P. Denatonium can also induce neurogenic inflammation, suggesting the involvement of functional bitter taste receptors. All these phenomena are dependent on TRPM5. Thus, across both the upper and lower respiratory tracts, there seems to be a conserved role for bitter taste receptors in protecting against bacterial infections.

#### 3.1.3. Gingival SCCs

The oral mucosa comprises the buccal mucosa, lingual mucosa, hard and soft palatal mucosa, and the gingiva. In addition to the type II TRCs in the taste buds on the surface of the tongue and soft palate, separate tuft cell-like structures were discovered in the gingival epithelium [39]. Gingival SCCs function as immune sentinels for periodontitis. Using bitter taste receptors and their corresponding downstream effectors, gingival SCCs recognize the quorum-sensing compound acyl-homoserine lactone (AHL), which is derived from the oral microbiome. Gnat3 ablation leads to severe alveolar bone loss upon gingiva ligature, an altered oral microbiome profile, and imbalanced expression of inflammatory and anti-inflammatory cytokines. Finally, the bitter tastant denatonium was found to slow the progression of periodontitis, suggesting novel strategies for preventing periodontal disease.

#### 3.1.4. Thymic Tuft Cells

The thymus is responsible for the maturation of T lymphocytes. In the thymic medulla, thymic epithelial cells present antigens from peripheral tissues, mediating the development of central tolerance. Recently, scRNA-seq was used to explore the cellular diversity of thymic epithelial cells [49,50]. A minor population of thymic epithelial cells was discovered that exhibited a gene expression profile like that of peripheral tuft cells, including the expression of IL-25 and components of the canonical taste transduction pathway. Mice lacking Pou2f3, the master tuft cell regulator, lack these thymic tuft cells and produce fewer NKT2 cells, which are the cellular sources of IL-4. Given that IL-25 and IL-4 are crucial mediators of the ILC2 circuit, thymic tuft cells seem to contribute to the microenvironment necessary for immune system regulation. Thus, although thymic tuft cells do seem to promote immune tolerance for tuft cell-restricted antigens, it remains unclear whether or how bitter taste receptors contribute to this process [49]. If bitter taste receptors are involved, pharmacologic activation with bitter tastants or inhibition with AITC could prove useful as novel therapeutic strategies for immune regulation.

#### 3.1.5. Urinary Tuft Cells

Since tuft cells in the intestine and airway secrete acetylcholine as a paracrine neurotransmitter, transgenic mice expressing eGFP in choline acetyltransferase (Chat)-expressing cells were used in the hunt for urethral tuft cells [51]. Urethral tuft cells adopt the canonical taste transduction cascade, including taste receptors, PLCβ_2_, and TRPM5. In addition, individual urethral tuft cells exhibit polymodal receptivity, responding to both sweet and bitter compounds. Application of tastants induced bladder detrusor activity, but this was blocked by co-application of the nicotinic receptor antagonist mecamylamine. Thus, urinary taste receptors and their downstream cascade seem to have a protective function in monitoring the contents of the urethral lumen.

#### 3.1.6. Neoplastic Tuft Cells

Tuft cells appear in both normal and neoplastic tissues. Inevitably, cancer disturbs the structure and balanced heterogeneity of cellular composition in normal tissues as a single cell type expands and crowds out its neighbors. One research group proposed that intestinal tuft cells act as cancer stem cells [63], and it is possible that a similar process occurs in some cases of respiratory cancer. Small cell lung cancers (SCLCs) can be divided into biological variants according to their expression of biomarkers such as ASCL1, NEUROD1, YAP1, and POU2F3 [64]. While ASCL1^high^ SCLCs show neuroendocrine-related profiles, the less common POU2F3^high^ SCLCs often exhibit a tuft cell-like expression signature that includes SOX9, GFI1B, ASCL2, and TRPM5. Given that pulmonary and bronchial tuft cells express several taste receptors, it is possible that the abnormally expanded tuft cells in these tumors also express taste receptors. If so, tastants could prove useful in the clinic for cancer treatment, but this remains to be proven.

### 3.2. Genuine-Ectopically Expressed Taste Receptors

#### 3.2.1. Enteroendocrine Cells

In the intestinal epithelium, enteroendocrine cells (EECs) detect various luminal nutrients to generate interoceptive hunger/satiety signals and modulate metabolic responses by secreting neuropeptides/hormones such as glucagon-like peptide-1 (GLP-1), cholecystokinin (CCK), serotonin (5-HT), etc. [65]. This nutrient-sensing capacity of EECs suggests the existence of EEC nutrient sensors. In this context, sweet and umami taste receptors are attractive candidates because they are already used for nutrient detection in TRCs in the mouth [66,67]. In exploring this hypothesis, several groups have employed the EEC-like cell line STC-1 [67,68,69]. Multiple groups tried to demonstrate the implication from in vitro experiments in animal models [70,71] and clinical studies [72,73]; however, most of them failed to lead to any meaningful results. For example, consumption or intragastric infusion of artificial sweeteners rather than natural sugars did not reach to increase the significant level of incretin hormones of EECs [70,71,72,73].

Recent single-cell RNA transcriptomic experiments have also cast doubt on the above EEC sweet taste receptor hypothesis. Multiple single-cell RNA sequencing experiments on whole intestinal epithelial cells indicated that, although Tas1r3 is expressed, its expression seems to be restricted to the tuft cells rather than the EECs [43]. In contrast, after sorting CCK-GFP-labeled cells and subjecting them to qRT-PCR at the single-cell level, roughly 20% of CCK-positive cells express significant levels of Tas1r3 but not Tas1r2 [74]. Given that intestinal tuft cells and EECs arise from common secretory cell progenitors [41,75], this discrepancy may be due to the early expression of CCK in undifferentiated cells exhibiting an expression profile somewhere between intestinal tuft cells and EECs. Independent studies using in vivo nerve imaging revealed that gut-infused sugars are detected by sodium-glucose cotransporter 1 (SGLT1) rather than Tas1r3 [74,76]. Instead, Tas1r3 only mediates cellular responses to artificial non-nutritive sweeteners, indicating a distinct role for sweet taste receptors in distinguishing tastants rather than nutrients [74].

#### 3.2.2. Pancreatic β-Cells

Pancreatic β-cells regulate systemic glucose levels by secreting insulin. Although pancreatic β-cells must obviously be able to detect changes in circulating glucose, pancreatic β-cells also release insulin in response to other nutrients such as fructose, as well as to artificial sweeteners such as saccharine [77]. Fructose evokes Ca^2+^ responses and insulin release in mouse pancreatic islets. Injections of mice with fructose induce or potentiate insulin secretion when paired with or without glucose, respectively. Interestingly, all these phenomena are abolished with genetic ablation of Tas1r2 or Tas1r3. Similarly, both pharmacologic inhibition of PLC signaling and genetic ablation of TRPM5 were found to impair fructose-induced insulin secretion, demonstrating the involvement of canonical sweet taste receptor signaling.

Nakagawa et al. reported the expression of sweet taste receptors in pancreatic β-cells and investigated their intracellular responses in detail [78]. Using mouse pancreatic islets, which included β-cells, as well as the glucose-responsive β-cell line MIN6, they found that sucralose induces an immediate and sustained elevation of cytoplasmic Ca^2+^ and cAMP simultaneously. The sweet taste receptors expressed in pancreatic β-cells seem to couple to a distinct signaling system that activates both the Ca^2+^ and cAMP second messenger systems. Since pharmacologic agents that increase cAMP production in β-cells would theoretically protect these cells from various stresses and from apoptosis, sweet taste receptors are potential targets in the hunt for novel diabetes treatments.

#### 3.2.3. Brain

The brain must not only process sensory information derived from peripheral sensory organs regarding nutrients, but it must also have mechanisms for responding to circulating nutrient levels to help it maintain systemic homeostasis. Glucose-sensing neurons in the brain are the most well-studied example. While some glucose-responsive neurons are excited by glucose, some are inhibited. Although the major molecular components for glucose detection in glucose-sensing neurons are similar to those in pancreatic β-cells [79], some glucose-sensing neurons also express sweet taste receptors [80]. This possibility was raised initially after a physiologic analysis of hypothalamic glucose-sensitive neurons. When the artificial sweetener sucralose was applied to ex vivo hypothalamic sections, specific hypothalamic neurons were activated. Pharmacologic inhibition of sweet taste receptors with the Tas1r3-specific antagonist gurmarin [80] alleviated these sucralose-induced responses, implying a role for sweet taste receptors in this phenomenon. Although RT-PCR experiments have also supported the brain expression of sweet taste receptors, the specific cell types responsible remain unknown. Recently, our own group generated Tas1r2-Cre mice and used them to demonstrate the expression of Tas1r2 across the brain in various regions [81]. Consistent with previous physiologic data [80], we observed Tas1r2 expression in the hypothalamic proopiomelanocortin (POMC)-expressing neurons whose activity indicates energy excess across the body. Interestingly, we also found Tas1r2 expression in the agout-related peptide (AgRP)-expressing neurons that oppose the function of the POMC neurons and whose activity indicates energy depletion across the body [81]. In addition to these hypothalamic neurons, we unexpectedly found expression of Tas1r2 across the entire brain. Tas1r2 expression was particularly high in the neuronal and non-neuronal cell populations comprising the circumventricular organs, such as the organum vasculosum lamina terminalis, the subfornical organ, the subcommisural organ, the median eminance (ME), and the area postrema. Expression in tanycytes and perivascular cells that line the wall of the third ventricle and ME, respectively, should be particularly advantageous for the simultaneous detection of various ligands in the cerebrospinal fluid (CSF) and blood. Thereby, Tas1r2 might be responsible for communication between CSF and blood in which the brain–blood barrier is loose. We also observed expression in vascular structures of the striatum and cerebral cortex [81]. Similarly, expression of other taste receptors such as Tas1r3, the binding partner of Tas1r2, and Tas2rs was detected in several brain areas, but the exact cell type expressing them was not mapped in detail [82,83]. Future studies will be necessary to clarify the roles these taste receptors play in the regulation of brain function.

#### 3.2.4. Urinary Bladder

In addition to the urinary tuft cells discussed earlier [51], there are other cellular components of the urinary tract that express taste receptors [84]. Umbrella cells, which reside in the most superficial urothelial layer, express Tas1r2 and Tas1r3 in both humans and rodents. Artificial sweeteners such as saccharin increased rat bladder smooth muscle contraction in strips of bladder tissue in response to electrical field stimulation only when the strip included the urothelium. In contrast to sweet taste receptors [84], the detrusor smooth muscle (DSM) expresses bitter taste receptors [85]. The bitter-tasting compound chloroquine (CLQ) induced a concentration-dependent relaxation of carbachol- and KCl-induced contractions in human DSM strips. Furthermore, higher concentrations of CLQ significantly inhibited spontaneous and electrical field stimulation-induced contraction. Different bitter compounds such as denatonium and quinine behaved similarly to CLQ. Taste receptor agonists have thus been suggested as candidate treatments for overactive bladder [85]. The exact mechanism underlying this phenomenon, however, should be investigated further. It remains unclear whether the pharmacologic effects of tastants on the relaxation of urinary smooth muscle originate in urethral tuft cells, umbrella cells, or in the detrusor smooth muscle itself.

#### 3.2.5. Heart

The subunits for the umami receptors showed some expression in cardiac fibroblasts, while more bitter taste receptor expression was found in cardiomyocytes [86]. Interestingly, there was some plasticity observed in these expression levels. For example, the expression of these receptors was developmentally regulated in the postnatal period. Moreover, under nutrient deprivation, several bitter taste receptors were found to be up-regulated both in vivo and in vitro. Although the details remain unclear, it does seem taste receptors have some physiologic relevance as metabolite/nutrient sensors in the heart.

The same research group also tried screening for bitter ligands capable of modifying cardiac function via bitter taste receptors expressed in cardiac tissue [87]. Using an in vitro Ca^2+^ mobilization assay on cells individually transfected with five bitter taste receptors cloned from rat heart, they tested 12 candidate bitter compounds from a panel of 102 natural and synthetic bitter compounds. In the Langendorff perfused-heart experiments, two representative bitter compounds, sodium thiocyanate for Tas2r108 and sodium benzoate for Tas2r137, exhibited concentration-dependent cardiac effects. Sodium thiocyanate and sodium benzoate led to a 40% decrease in left ventricular developed pressure and an increase in aortic pressure, respectively. Changes in cardiac function induced by bitter compounds were inhibited by the Gαι inhibitor pertussis toxin or the Gβγ inhibitor gallein, implicating G-protein signaling. Consistent with these results, independent research groups also found that the bitter compound denatonium similarly reduced cardiac contractility [87].

#### 3.2.6. Respiratory Smooth Muscle

Pan-GPCR screening in human airway smooth muscles revealed abundant expression of several bitter taste receptors compared to the also prominent expression of β-adrenoreceptor ADRB2 [88]. Bitter compounds induced intracellular Ca^2+^ responses that were dependent on Gβγ, PLCβ, and IP_3_ receptor signaling. Compared to isoproterenol, which is the β-adrenergic agonist and the current gold standard drug for bronchoconstriction, bitter compounds showed more superior potency in evoking relaxation of airway smooth muscle in vitro and tracheal tension in vivo [88,89,90]. Finally, bitter compounds administered in an aerosol form alleviated the asthmatic symptoms of mouse models for allergic airway inflammation and bronchial hyperresponsiveness. The same group performed a follow-up study that used siRNA knockdown in airway smooth muscle cells to clarify that the signaling pathway responsible for this phenomenon required G*ai*2 rather than GNAT3 [91]. However, other groups also demonstrated the roles of the canonical taste receptor signaling pathway in alleviating L-type Ca^2+^ channel, which mediates bronchoconstriction [92]. Despite the contradictory explanation for the mechanism [91,92], powerful potency recapitulated by independent research groups and anatomical location of bronchus easy to administer imply the feasibility of therapeutic potentials of taste receptors and their ligands.

#### 3.2.7. Vascular Smooth Muscle

Like bronchial smooth muscle, vascular smooth muscle also expresses several bitter taste receptors [93]. Their expression levels are similar to that of the α1A adrenoceptor, which is critical for the contraction of smooth muscle. Bitter tastant application induced muscle relaxation in a manner antagonistic to the α1A adrenoceptor. This relaxation-inducing action of bitter tastants on smooth muscle was found to be independent of the endothelium, indicating a cell-autonomous role for the bitter taste receptors in smooth muscle cells.

#### 3.2.8. Respiratory Ciliated Cells

In the airway epithelium, most epithelial cells are ciliated. Motile cilia beat together to move luminal irritants out of the airway. This beating of these motile cilia is regulated by taste receptor signaling [94]. Single ciliated epithelial cells in the human airway were found via immunostaining with specific antibodies to express multiple bitter taste receptors. Interestingly, the subcellular localizations of the different receptors were distinguishable from one another, with TAS2R4 and TAS2R38 localizing near the ciliary tip and TAS2R43 and TAS2R46 localizing to the ciliary base. Application of several different bitter compounds not only elicited intracellular Ca^2+^ responses but also accelerated ciliary beat frequency via nitric oxide (NO) synthesis. In more recent scRNA-seq data [47,48], however, taste receptors and their downstream effectors were only observed in airway tuft cells. This means that there seems to be a discrepancy between the expression of bitter taste receptors at the RNA and protein levels.

#### 3.2.9. Immune Cells

In the purified human blood cells, chemosensory receptors showed cell type-specific expression patterns [95]. For example, most ORs were strongly biased in B cells, T cells, and polymorphonuclear neutrophils (PMNs), while a few specific ORs were detected in monocytes and natural killer (NK) cells. Compared to ORs, taste receptors showed weakly biased expression patterns. Using antibodies against ORs and taste receptors, fluorescent-associated cell sorting and immunocytochemistry revealed that chemosensory receptors are co-expressed in the subpopulations of dedicated cell types. Exposure to either sweet or bitter tastant induced PMNs migration, which was blocked by Gi inhibitor, or gustducin knockdown, indicating the functional expression of taste receptors in PMNs.

Given that polymorphism of TAS2R38 is associated with multiple immune-related diseases [96,97,98,99], TAS2R38 in immune cells has been paid attention to. TAS2R38 was higher expressed in CD3^+^ T cells than CD19^+^ B cells and in CD4^+^ T cells than in CD8^+^ T cells [100]. As T cells become mature, TAS2R38 expression increases. Application of goitrin, TAS2R38 activator, induced calcium flux in T cells and tumor necrosis factor- α (TNF-α). According to the polymorphism type of TAS2R38, the response of T cells to goitrin differed, implying the functional contribution of TAS2R38 in T cells. Later, other groups also agreed with TAS2R38 expression in lymphocytes, which were infiltrated in AD skin, but they argued against the roles of TAS2R38 in lymphocytes [101]. Activation of TAS2R38 with agonists rather inhibited cell migration and unexpectedly suppressed TNF-α production regardless of TAS2R38, which implies the off-target effect of the chemicals.

In macrophages, the signal transduction mechanism of bitter taste receptors was more extensively investigated [102]. Fluorescent imaging to measure the intracellular secondary messengers and pharmacologic manipulation revealed a sequential signaling cascade induced by the activation of bitter taste receptors with bitter tastants or microbe-derived LasI, which increases intracellular Ca^2+^ and produces NO and cyclic guanosine monophosphate (cGMP). Independently, downregulation of the cAMP level was also observed, but it is irrelevant to NO and cGMP signaling. Finally, both pathways resulted in the phagocytosis of macrophages. Furthermore, in vitro co-culture of macrophage and H441 cells, the airway cell lines, recapitulated the cross-reactivity of macrophage to NO diffused from upper airway ciliated cells, facilitating phagocytosis.

As immune cells communicate with various types of cells via cytokines, the effects of taste receptors in immune cells should be considered systemically together with those of taste receptors in the communicating partner cells.

#### 3.2.10. Adipose Tissue

Adipocytes also express taste receptors, but their roles in adipogenesis remain controversial [103,104]. Genetic loss of taste-related genes, such as Tas1r2, Tas1r3, or GNAT3, was found to reduce adipose tissue size when mice were fed a high-fat diet [105,106]. Both murine and human adipocytes express the taste receptors Tas1r2, Tas1r3, and TAS2R38. In vitro application of artificial sweeteners or bitter tastants was found to affect the differentiation of 3T3-L1 cells and embryonic mesenchymal stem cells into mature adipocytes [103,104,106]. 3T3-L1 cells treated with tastants exhibited increases in intracellular Ca^2+^ and cAMP [104], as well as activation of the Akt-ERT1/2 pathways [103]. This led to the differentiation of preadipocyte-like cells into mature adipocytes. Based on the above evidence, one group concluded that taste receptors in preadipocytes are crucial in the regulation of adipogenesis. Other groups, however, reported that Tas1r2 and Tas1r3 double KO mutants showed comparable levels of tastant-induced adipogenesis and Akt phosphorylation, suggesting that these two receptors at least are dispensable [103]. This means that just because cells exhibit ectopic taste receptor expression does not always mean those taste receptors play a significant role in the physiology of those cells. We must also consider the possibility that tastants activate some unknown off-target receptor rather than the taste receptors themselves.

## 4. Conclusions

Here, we have summarized the literature regarding the ectopic expression of taste receptors (Table 1). Ectopic taste receptors are largely involved in three biological processes. First, taste receptors can help regulate metabolism via their ability to detect circulating nutrients and exogenous compounds. This is the function of the intestinal EECs, pancreatic β-cells, and hypothalamic neurons. Second, upon exposure to microbe-derived compounds, taste receptors expressed in tuft cells can trigger immune responses in communication with immune cells via the ILC-2 circuit. Last, taste receptors help tune involuntary muscle tone.

How are ectopic taste receptors activated? Tuft cells are exposed to the lumen of the digestive and respiratory tracts, giving them direct access to taste receptor ligands. If the ectopic taste receptors are localized deep within a particular tissue, however, the physicochemical and pharmacokinetic properties of each exogenous compound would determine whether it can act as a ligand for ectopic taste receptors in vivo. Alternatively, ectopic taste receptors deep in the tissues may also respond to unknown endogenous ligands.

Conventional tastants are recognized not only by taste receptors but also by off-target receptors. As most of the studies of the function of ectopic taste receptors have relied on the pharmacologic application of tastants in vitro and in vivo, it can be easy to misattribute observed effects to a contribution of the ectopic taste receptors themselves. Therefore, genetic approaches should always accompany pharmacologic approaches to validate the functions of ectopic taste receptors. We do expect, however, that further investigations will confirm at least some of the perceived therapeutic potential of taste receptors and their ligands.

## Figures and Tables

**Table 1 ijms-23-09677-t001:** Overview of expression pattern and function of ectopic taste receptors.

Organ	Cell Type	Receptor	Function	Reference
Intestine	Tuft cell	Tas1r3	Regulation of epithelial homeostasis	[61]
		Tas2rs	Sensing the microbe-derived molecules and secretes IL-25	[59]
	EECs	Tas1r3	Mediating cellular responses to artificial non-nutritive sweeteners	[74]
Respiratory tracts	Tuft cell	Tas2rs	Release of acetylcholine to activate nociceptive neurons	[46,62]
		Tas1r3	Alleviation of bitter tastants-evoked responses upon glucose stimulation	[45]
	Smooth muscle	Tas2rs	Inducing intracellular Ca^2+^ responses Relaxation of airway smooth muscles and tracheal tension Alleviation of the asthmatic symptoms of mouse models for allergic airway inflammation and bronchial hyperresponsiveness	[88,89,90,91,92]
	Ciliated cell	TAS2Rs	Eliciting intracellular Ca^2+^ responses and accelerating ciliary beat frequency	[94]
Gingiva	SCCs	Tas2rs	Alleviation of periodontitis	[39]
Thymus	Tuft cell	Tas2rs	Establishment of the microenvironment necessary for immune cells maturation	[49]
Urinary tract	Tuft cell	Tas1r1, Tas1r3, Tas2rs	Monitoring the contents of the urethral lumen	[51]
	Umbrella cell	Tas1r2, Tas1r3	Contract bladder smooth muscle	[84]
		Tas2rs	Relaxation of smooth muscle contractions	[85]
Pancreas	β-cells	Tas1r2, Tas1r3	Induction and potentiation of insulin secretion	[77,78]
Brain	Neurons	Tas1r2	Eliciting intracellular Ca^2+^ responses to artificial sweeteners	[80]
	Tanycytes, perivascular cells	Tas1r2	Eliciting intracellular Ca^2+^ responses to artificial sweeteners in tanycytes	[81]
	Unknown	Tas2rs	Unknown	[82,83]
Heart	Fibroblast	Tas1r1, Tas1r3	Unknown	[86]
	Cardiomyocyte	Tas2rs	Regulation of cardiac and aortic pressure	[86,87]
Vascular systems	Smooth muscle	Tas2rs	Relaxation of vascular smooth muscles	[93]
Immune cells	T cells	Tas1rs Tas2rs	Cell migration Secretion of TNF-α	[95]
				[100]
	Macrophages	Tas1rs Tas2rs	Chemotaxis Facilitation of phagocytosis	[95]
				[102]
Adipose tissue	Preadipocyte	Tas1r2, Tas1r3	Differentiation of preadipocyte-like cells into mature adipocytes	[104]
		TAS2R38	Unknown	[106]

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
