# Peer review of "Taste Receptors beyond Taste Buds"

_ijms, 2022, doi:10.3390/ijms23179677_

Round 1
Reviewer 1 Report
The presence of extra oral taste receptors has been the subject of intense investigation over the last 20 years and there have been several reviews on the topic, even recently (2021-2022), see below:
D'Urso O, Drago F. Pharmacological significance of extra-oral taste receptors. Eur J Pharmacol. 2021 Nov 5;910:174480. doi: 10.1016/j.ejphar.2021.174480. Epub 2021 Sep 5. PMID: 34496302.
Medapati MR, Bhagirath AY, Singh N, Chelikani P. Pharmacology of T2R Mediated Host-Microbe Interactions. Handb Exp Pharmacol. 2022;275:177-202. doi: 10.1007/164_2021_435. PMID: 33580389.
Sharma P, Conaway S Jr, Deshpande D. Bitter Taste Receptors in the Airway Cells Functions. Handb Exp Pharmacol. 2022;275:203-227. doi: 10.1007/164_2021_436
Tuzim K, Korolczuk A. An update on extra-oral bitter taste receptors. J Transl Med. 2021 Oct 21;19(1):440. doi: 10.1186/s12967-021-03067-y
Wang H, Matsumoto I, Jiang P. Immune Regulatory Roles of Cells Expressing Taste Signaling Elements in Nongustatory Tissues. Handb Exp Pharmacol. 2022;275:271-293. doi: 10.1007/164_2021_468
Behrens M, Lang T. Extra-Oral Taste Receptors-Function, Disease, and Perspectives. Front Nutr. 2022 Apr 4;9:881177. doi: 10.3389/fnut.2022.881177
The content of the current review presented by Ki and Jeong is, overall, rather descriptive. I do not think it brings significantly different information than the aforementioned recent reviews. Nevertheless, here are some edits and suggestions to improve its impact.
Importantly, the authors should try to emphasize even more on the potential functional significance of the extra-oral expression of these receptors. As an example, why would there be sweet taste receptors in the bladder (umbrella cells exposed to urine)? What would be the potential sweet tasting ligands present in the urine? Glucose, that typically filters in higher amounts into the urine upon kidney failure or diabetes, for example, should not be there at concentrations sufficient to activate the sweet taste receptor (Glucose EC50 is >> 100 mM for the receptor), what other ligands then? What are the conclusions that can be drawn from studies with taste receptor k/o or taste signaling k/o strains? What else could/should be done? Are there any differences under normal or pathological states? Author should critically assess what is missing from the current data and what are the next steps. These types of questions should be asked for every section.
On page 5 section 3.2.1, authors need to explain the “EEC sweet taste receptor hypothesis”: I presume they are referring to the hypothesis that the gut sensing mechanism responsible for the incretin effect (greater insulin release observed when providing glucose through the GI rather than when injected IV) is activation of the sweet taste receptor expressed in gut EECs… The authors raise doubts on the hypothesis based on expression data. Of note, the initial results reported from isolated cells or in animal studies did not reproduce in later studies (see Fujita et al and Glendinning et al., below). Infusion of nonnutritive sweeteners did not lead to an increase in GLP-1 or GIP levels in rats and sub-chronic administration of nonnutritive sweeteners caused only limited metabolic effects in mice . However, more importantly, there has been a plethora of human studies showing that consumption or intragastric infusion of high levels of high potency nonnutritive sweeteners, capable of robustly activating the human sweet taste receptor, does not lead to secretion of GLP-1, GIP and ultimately insulin. Similarly, these sweeteners do not lead to a further increase of the secretion of these molecules in combination with glucose (see articles below). The authors should point to these studies and also revise their conclusion (lines 392 to 399). It is in fact unlikely, based on a preponderance of human and clinical trial studies, that expression of the sweet taste receptor in EECs or pancreatic b-cells regulate metabolism.
Ford HE, Peters V, Martin NM, Sleeth ML, Ghatei MA, Frost GS, Bloom SR. Effects of oral ingestion of sucralose on gut hormone response and appetite in healthy normal-weight subjects. Eur J Clin Nutr. 2011 Apr;65(4):508-13. doi: 10.1038/ejcn.2010.291
Wu T, Zhao BR, Bound MJ, Checklin HL, Bellon M, Little TJ, Young RL, Jones KL, Horowitz M, Rayner CK. Effects of different sweet preloads on incretin hormone secretion, gastric emptying, and postprandial glycemia in healthy humans. Am J Clin Nutr. 2012 Jan;95(1):78-83. doi: 10.3945/ajcn.111.021543
Steinert RE, Frey F, Töpfer A, Drewe J, Beglinger C. Effects of carbohydrate sugars and artificial sweeteners on appetite and the secretion of gastrointestinal satiety peptides. Br J Nutr. 2011 May;105(9):1320-8. doi: 10.1017/S000711451000512X
Maki KC, Curry LL, Reeves MS, Toth PD, McKenney JM, Farmer MV, Schwartz SL, Lubin BC, Boileau AC, Dicklin MR, Carakostas MC, Tarka SM. Chronic consumption of rebaudioside A, a steviol glycoside, in men and women with type 2 diabetes mellitus. Food Chem Toxicol. 2008 Jul;46 Suppl 7:S47-53. doi: 10.1016/j.fct.2008.05.007
Ma J, Bellon M, Wishart JM, Young R, Blackshaw LA, Jones KL, Horowitz M, Rayner CK. Effect of the artificial sweetener, sucralose, on gastric emptying and incretin hormone release in healthy subjects. Am J Physiol Gastrointest Liver Physiol. 2009 Apr;296(4):G735-9. doi: 10.1152/ajpgi.90708.2008
Fujita Y, Wideman RD, Speck M, Asadi A, King DS, Webber TD, Haneda M, Kieffer TJ. Incretin release from gut is acutely enhanced by sugar but not by sweeteners in vivo. Am J Physiol Endocrinol Metab. 2009 Mar;296(3):E473-9. doi: 10.1152/ajpendo.90636.2008
Glendinning JI, Hart S, Lee H, Maleh J, Ortiz G, Ryu YS, Sanchez A, Shelling S, Williams N. Low-calorie sweeteners cause only limited metabolic effects in mice. Am J Physiol Regul Integr Comp Physiol. 2020 Jan 1;318(1):R70-R80. doi: 10.1152/ajpregu.00245.2019
Section 3.2.6 Respiratory smooth muscle. I think that this section is actually underrepresented. Much more work on validation has been done with respect to the potential effect of bitter taste receptors on bronchodilation then, let’s say, taste receptors in the bladder, but they both get the same level of attention. The potential use of bitter receptor agonists for asthma therapy is actually very appealing. Please see additional work by Deepak A Deshpande and Stephen B Liggett and further expand. In addition, I suggest replacing the potential effect of taste receptors on metabolism with their effect on lung function, in the conclusion section. Maybe this is what the authors are referring to with “tune involuntary muscle tone”. Still, much more taste receptor investigative work has been done on bronchial vascular smooth muscle cells than any other models (heart, bladder)….
Conclusion section should be toned down. It is written as if there is no question that extraoral taste receptors are involved in metabolism regulation, are involved in innate immunity responses and are involved in controlling involuntary muscle tone. While the progress made over the last several years is undeniable, to get to this level of certainty we actually need more translational research and even small molecule pharmacological tools, altering the functions of these receptors, and showing the predicted effect in clinical trials…
Finally, the section 2.2 on signal transduction. The phrase starting with…Then… on line 58 and ending with GNAT3 on line 61 is not clear and is not accurate. One did not have to use promiscuous G proteins to identify Gb3, Gg13 or a-gustducin as key players for taste signaling. The authors should define which G proteins typically couples to taste receptor (probably Gi proteins in general, including gustducin) and the downstream effectors.
Reviewer 2 Report
Young et al reviews the expression and potential roles of Tas1rs and Tas2rs in extra-oral tissues. However, the article is not a comprehensive resource of the documented expression of these taste receptor genes in non-taste cells, as the title would suggest. In many instances, the authors skim through each tissue or cell the authors without describing the roles of the receptors in any detail. There is also no mention of the expression of Tas2rs and Tas1rs in immune cells (macrophages T and B cells etc.), brain and skin, and of T1rs in tanycytes and other brain cells. The paper will also benefit from a table outlining the cell types and receptors known to be expressed in them, and/or figures. It is not clear what is meant by the subtitle 3.2. ‘Genuine-ectopically expressed taste receptors’ and how the authors differentiate it from section 3.1 which is exclusively about the receptors in immune surveillance roles. It is better to organize the manuscript as they summarize in the first paragraph of the conclusions section. of Reference 5 is about discovery of Tas2rs only, not of T1r2 and T1r1 as implied in Section 2.1.
Section 2.2 should do a better job of describing how the members of the downstream signal transduction machinery were discovered. Currently, they imply ‘in vitro heterologous expression of taste receptors and a promiscuous Gα protein’ was the main method, while most of them used screening of taste cDNA libraries to identify these genes in taste cells. Further, the authors should clarify that they mean Tas1rs and Tas2rs only, when they state ‘taste receptors’, since there is no discussion of salty and sour taste receptors.
Round 2
Reviewer 1 Report
thank you for answers to comments and suggestions
Reviewer 2 Report
The authors have addressed all my concerns in the revised version of the manuscript.